# High-Throughput Proteomic Profiling of Nipple Aspirate Fluid from Breast Cancer Patients Compared with Non-Cancer Controls: A Step Closer to Clinical Feasibility

**DOI:** 10.3390/jcm10112243

**Published:** 2021-05-21

**Authors:** Amy L. George, Sadr ul Shaheed, Chris W. Sutton

**Affiliations:** Institute of Cancer Therapeutics, University of Bradford, Tumbling Hill Street, Bradford BD7 1DB, UK; alg76@medschl.cam.ac.uk (A.L.G.); sadr.shaheed@nds.ox.ac.uk (S.u.S.)

**Keywords:** breast cancer, biomarkers, proteomics, nipple aspirate fluid, liquid biopsy, non-invasive

## Abstract

Background: Early detection of breast cancer (BC) is critical for increasing survival rates. However, current imaging approaches can provide ambiguous results, requiring invasive tissue biopsy for a definitive diagnosis. Multi-dimensional mass spectrometric analysis has highlighted the invaluable potential of nipple aspirate fluid (NAF) as a non-invasive source of early detection biomarkers, by identifying a multitude of proteins representative of the changing breast microenvironment. However, technical challenges with biomarker validation in large cohorts remain due to low sample throughput, impeding progress towards clinical utility. Rather, by employing a high-throughput method, that is more practicable for clinical utility, perturbations of the most abundant NAF proteins in BC patients compared with non-cancer (NC) controls could be monitored and validated in larger groups. Method: We characterized matched NAF pairs from BC (*n* = 9) and NC (*n* = 4) volunteers, using a rapid one dimensional liquid chromatography-mass spectrometry (1D LC-MS/MS) approach. Results: Overall, 198 proteins were relatively quantified, of which 40 were significantly differentiated in BC samples, compared with NC (*p* ≤ 0.05), with 26 upregulated and 14 downregulated. An imbalance in immune response and proteins regulating cell growth, maintenance and communication were identified. Conclusions: Our findings show 1D LC-MS/MS can quantify changes reflected in the NAF proteome associated with breast cancer development.

## 1. Introduction

Breast cancer (BC) is the most common cancer diagnosed among women worldwide and impacts 2.1 million females each year. In 2020, it was estimated that 684,996 women died from the disease, and it remains the principal cause of cancer-related deaths in women in more than 100 countries [1]. Individuals diagnosed at an advanced stage have a significantly reduced five-year survival rate, ranging from 10–50%, as compared with the significantly higher survival rate for earlier detected, more localized tumors, at approximately 90% [2,3]. These trends may reflect the availability of national screening programs, later diagnosis, and poor access to treatment in some countries, and importantly highlight breast cancer as a tremendous public health problem. Continuous global improvement in diagnostic tests for early-stage breast cancer is therefore essential to provide patients with enough lead time for optimized prevention and treatment, in both industrialized countries and low-resource areas.

Suspected breast tumors are traditionally identified using a combination of imaging techniques, each with unique advantages and limitations in respect to resolution, sensitivity, and contrast generation [4]. Within the UK, the NHS breast cancer screening program is available for women between the ages of 47 and 72 to undergo mammography every three years. A 20% reduction in BC-related deaths has been attributed to its implementation but it is also associated with a high rate of false-positive diagnosis, leading to prolonged patient anxiety during re-screening, and costly and invasive follow-up tests [5]. Ultimately a solid tissue biopsy is taken for histopathology to provide a definitive diagnosis, of which a recent study showed 93.8% of cases proved benign. This means that for a single breast cancer case identified by mammography, 15 women without cancer were subjected to unnecessary invasive procedures [6]. Moreover, as an independent diagnostic tool for early-stage disease or aggressive malignancies, imaging techniques are generally limited as they have a reduced threshold of detection in dense breasts and are less sensitive to small tumors [7]. Efforts to improve such limitations, with the development of digital breast tomosynthesis, have shown to reduce the number of benign biopsies performed at screening assessment. The technique showed higher specificity across different feature types, including small mass lesions and asymmetric densities, and improved the specificity of micro calcifications when compared with digital mammography [8]. While the technique is being explored in various countries as a diagnostic tool, it is not routinely available [9,10]. Hence, there remains a need for a new, non-invasive method of assessing breast health, complementing current procedures for the detection of pre-cancerous and cancerous breast lesions.

A promising alternative is the analysis of nipple aspirate fluid (NAF), a breast-specific proximal fluid that is secreted naturally by the ductal-lobular system in the breasts of non-lactating adult women [11]. NAF can be obtained with varying degrees of success (34–90%) from both pre- and post-menopausal women by gentle breast massage, using a milk-expressing pump or stimulated by oxytocin nasal spray [12,13,14,15,16]. It has a rich composition of proteins, hormones, lipids and carbohydrates with cellular debris shed from the ductal and lobular epithelium and is therefore considered to be a mirror of the cellular changes occurring in the breast microenvironment in both physiological and pathological conditions [17,18]. As the vast majority of breast cancer cases arise from the epithelial cells lining the ductal–lobular system, analysis of the fluid produced therein, as a source of biomarkers for precancerous and cancerous transformation, may provide a faster, cheaper means of assessing tumor dynamics without the need for invasive procedures [19,20]. In comparison to tissue biopsy the technique is much less burdensome as NAF is more accessible and considerably less invasive. Thus, there is the potential for longitudinal health assessment, with reduced risk of complications. Personalized disease perturbations may be monitored in combination with current imaging to detect the earliest signs of abnormalities, through molecular changes, allowing greater lead time for earlier intervention.

Proteomic technologies have been fundamental in facilitating the characterization of NAF protein profiles on a comprehensive level. Most studies have focused on mining the biological fluid using fractionation at the peptide level to reduce sample complexity and increase the depth of analysis. However, this approach has its drawbacks as there is a consequential increase in single sample-preparation and analytical steps, often leading to lengthy workflows taking multiple days or even weeks for complete analysis. Brunoro et al. demonstrated this when comparing efficiencies of reverse phase (RP)-30 cm, strong-cation exchange (SCX)/RP-10 cm, and OFFGEL/RP-10 cm peptide-centric sample preparation approaches which, in total, identified 193, 390 and 528 proteins, respectively. The OFFGEL pre-fractionation technique resulted in a larger number of identified peptides but suffered from longer analysis time (>1400 min) in comparison with the other two methods (RP-30 cm: 168 min; SCX/RP-10 cm: 468 min) [21]. When previously conducting an interrogation into the NAF proteome in both non-cancer (NC) and BC matched samples using SCX, our group also encountered a substantial increase in sample preparation and analysis time, although providing the most comprehensive multidimensional characterization of NAF to date by identifying 1990 proteins [22].

While multi-dimensional separation is currently the gold standard approach for biomarker discovery, advances in column technology have made one-dimensional liquid chromatography mass spectrometry (1D LC-MS/MS) a potential alternative as the high resolution can be retained without the additional separation steps, thus requiring less sample material and less mass spectrometry runtime, increasing throughput substantially as has been demonstrated for serum [23]. Since NAF is secreted and collected in nominal volumes, its analysis with minimalistic workflows is essential to reduce cumulative sample loss from sequential steps of the analysis.

In this pilot study, the aim was to assess the feasibility of using NAF as a liquid biopsy for the investigation of breast health using a more clinically adaptable one-dimensional workflow. To investigate whether the complexity of NAF samples could be reduced using nano-liquid chromatography coupled to an Orbitrap Fusion mass spectrometer, while providing sufficient resolution for significant protein identification, matched pairs from non-cancer and cancer cases were semi-quantitatively analyzed following in-solution digestion.

## 2. Experimental Section

### 2.1. Study Participants and Sample Collection

The study was approved by University of Bradford’s Independent Scientific Advisory Committee (reference: application/13/051), ethically approved by the NHS Leeds (East) Research Ethics Committee (reference: 07/H1306/98+5) and informed consent was obtained from 112 participants. NAF expression was induced by manual breast massage, either by the individual themselves (in the cases of disease-free volunteers) or by the clinician prior to surgery (with patients under general anesthetic). Where possible, aspiration of each breast was attempted to provide matched pairs. Samples were collected into tubes pre-treated with protease inhibitor cocktail mixture (Roche Diagnostics), frozen within 30 min of collection (−20 °C) and stored long term at −80 °C until required for analysis. A total of 168 NAF samples were collected comprising 56 pairs and 56 single, first expression liquid biopsies. Twenty-six paired NAF samples from 13 women were chosen for 1D LC-MS/MS analysis, based on defined eligibility criteria: (a) being matched samples, (b) having at least 200 μg of starting material required for our exploratory workflow and (c) containing lower levels of albumin (to avoid dominating proteomic analysis), identified by Coomassie brilliant blue staining of sodium dodecyl sulfate–polyacrylamide gel electrophoresis (SDS-PAGE). Clinical parameters of participants and physical properties of the samples are summarized in Table 1.

### 2.2. Sample Characterisation

NAF samples were characterized for volume and color before being diluted 1-in-20 or 1-in-50 with LCMS grade water, dependent on observable viscosity. Protein concentration of each sample was determined using the Bradford assay according to manufacturer’s guidelines, and 20 μg of protein was analyzed by SDS-PAGE followed by Coomassie blue staining.

### 2.3. Proteomic Analysis

NAF samples were processed in small batches, of typically 8 to 10, with the matched pair from subject 9 included with each group for measures of reproducibility and inter-experiment variation. Volumes of NAF, equivalent to 200 μg of protein, were concentrated by lyophilization at 48 °C under aqueous conditions and then solubilized in 20 μL of 8 M urea in 400 mM ammonium bicarbonate, 10% acetonitrile. Proteins were reduced with 4 μL of 50 mM dithiothreitol (DTT) at 60 °C for 30 min and alkylated using 4 μL of 100 mM iodoacetamide (IAA) at room temperature in the dark for 30 min. Urea concentration was then diluted from 8 mM to 2 mM with 52 μL of 400 mM ammonium bicarbonate in 10% acetonitrile, creating an optimal environment for proteolytic digestion. MS-grade trypsin (Fisher Scientific, Loughborough, UK) was added at a protease-to-protein ratio of 1:10 (*w*/*w*) at 37 °C for 3 h. A second aliquot of trypsin was applied after 3 h (giving a final protease-to-protein ratio of 1:5) and incubated overnight with gentle agitation. Following digestion, each sample was desalted on an Isolute C18 desalting column (Biotage, Uppsala, Sweden), lyophilized as previous described and stored at −20 °C until analyzed.

Tryptic peptides were reconstituted in 2% acetonitrile with 0.1% formic acid (FA) in water to a concentration of 1 μg/μL and analyzed in triplicate using a nano-LC Ultimate 3000 RSLC system (Dionex, Sunnyvale, CA, USA) coupled to an Orbitrap Fusion™ Tribrid™ mass spectrometer (ThermoFisher Scientific, Germany). Peptides were initially loaded (2 μg) at a 25 μL/min flow rate to a 5 mm Acclaim™ PepMap™100 C_18_ NanoViper Trap column packed with 5 μm silica particles, 100 Å pore size, followed by separation at 300 nL/min on an Acclaim PepMap100™ NanoViper column (25 cm × 75 μm ID, 2 μm particle) with an elution gradient of 5–80% Solvent B at 40 °C. Peptides were eluted using a binary system of Solvent A (acetonitrile, 0.1% FA in water) and Solvent B (100% acetonitrile, 0.1% FA) over an 80-min gradient, followed by a wash and re-equilibration of the analytical column. The spray voltage was set to 1.85–2.1 kV with an ion transfer temperature of 275 °C and no sheath or auxiliary gas flow used. Data-dependent acquisition was performed on an Orbitrap Fusion™, in positive ion mode, acquiring full MS spectra in the Orbitrap analyzer at a 120,000 resolution, automatic gain control (AGC) value of 3 × 10^5^ and scan range set to 350–1500 m/z. Precursor ions with charged states of 2+ to 7+ were sequentially submitted to collision-induced dissociation fragmentation and MS^2^ spectra were acquired in the ion trap with a 3 s cycle time, using the following parameters: AGC target value of 2 × 10^4^, normalized collision energy of 35%, minimum signal threshold of 2000 counts.

### 2.4. Data Analysis

MS raw files were loaded on to Proteome Discoverer 2.2 (ThermoFisher, Germany) using Mascot software version 2.5 (Matrix Science, UK) and searched against the *Homo sapiens* SwissProt database version 2019 containing 562,118 entries. Search parameters were as follows: trypsin as the enzyme for protein digestion, with a maximum of 2 missed cleavage sites per peptide; a mass tolerance for precursor candidates of 10 ppm (MS) and 0.8Da for fragment peak matching (MS/MS). Dynamic modifications included oxidation of methionine and deamidation of asparagine and glutamine, and static modifications included carbamidomethylation of cysteine residues. Only Master Proteins (containing at least one unique peptide and ≥2 peptide spectrum matches) were accepted and were identified with high confidence peptides by applying target decoy database search parameters (strict false discovery rate (FDR) of 0.01 and relaxed 0.05 FDR). Only those identifications in the 95% confidence interval threshold (*p* ≤ 0.05, Mascot score ≥23) were accepted and included in analysis.

Label-free quantitation (sum of the peak areas of the three most abundant unique peptides per protein) was Log_2_ transformed and filtered for 70% data completeness in at least one experimental group, NC, or BC in Perseus version 1.6.12 [24]. Values were standardized between samples based on the population’s normal distribution, and remaining missing values replaced by drawing random samples from a normal distribution (downshifted mean by 1.8 standard deviation (SD) and scaled SD (0.3) relative to the proteome abundance distribution). This provided a final dataset of 198 protein groups, with which we performed the statistical analysis.

Statistical inference was conducted by paired two-sample *t* test for bilateral samples (disease and healthy breast). Unilateral disease samples were further tested against NC volunteer samples using an unpaired student *t*-test. An unpaired two sample student *t* test was used for comparing NC and BC groups. Data was visualized as heat maps and volcano plots with R-studio. Following comparative proteomic profiling and quantitative comparison between disease groups, enrichment of gene ontology (GO) and functional analysis of significant proteins was carried out using FunRich Version 3.1.3; Functional Enrichment Analysis Tool (www.funrich.org (accessed on 4 February 2021)) focusing on biological process and molecular function [25]. The protein–protein interactions of those differentially expressed were analyzed using STRING database version 11.0 (https://string-db.org/ (accessed on 10 February 2021)) and pathway analysis conducted using Reactome Pathway Database version 75.0 [26,27].

## 3. Results

### 3.1. NAF Characterisation

Most volunteers expressed NAF, yielding enough volumes of sample for proteomic analysis (13 μL to 198 μL, with a median volume of 62 μL) within the first attempt. Protein concentrations were consistent with most recent NAF reports (3.28 mg/mL to 81.2 mg/mL, with an average concentration of 36.77 mg/mL). The color and consistency of NAF varied amongst individuals, and between breasts of a matched pair (Table 1), as did their overall protein profiles indicated by SDS-PAGE (Appendix A). Many samples were brown, opaque, and of a waxy consistency which required dilution with 50 μL of water to allow for pipetting due to high viscosity. Other samples were white, translucent and of a watery consistency.

### 3.2. A One-Dimensional Insight into the NAF Proteome

In total, 198 proteins were quantified between NAF samples, with an average of 167 proteins per sample (SD ± 20) (Figure 1a). Raw intensity peak area values were filtered to include proteins with peak area data complete for 70% of samples in at least one experimental condition (NC or BC). Subsequent comparison of accession lists from each experimental group (NC or BC) revealed all proteins to be common between the two conditions and ultimately no exclusive proteins were identified (Figure 1b). When plotting Log_2_ transformed protein abundances of all proteins in biological replicates 1, 2, and 3 of subject 9, it is clear the method provides significantly high concordance among identically processed samples between experiments, indicated by positive Pearson’s correlation coefficients (R) (mean R = 0.711, *n* = 6). Technical replicates (triplicate injection of each sample for mass spectrometric analysis) also correlate highly (mean R = 0.987, *n* = 6), indicating robust instrument performance (Figure 1c—right breast, Appendix A—left breast).

Global Pearson correlation of all samples showed that protein profiles of matched pairs from the same individual (normal vs. disease) had a strong and significant positive correlation in all cases, consistent with our previous investigations in suggesting matched, paired proteomes to be most alike (Figure 1d) [22]. Differential analysis was performed between intra-individual paired NAF samples from unilateral breast cancer patients (using the contralateral non-diseased breast sample as negative control). A paired student *t*-test between bilateral samples revealed significant differences in 22 proteins (Log_2_ fold change −0.8 to −1.8), discriminating disease breast from healthy within an individual (Figure 1e) (Appendix A). Unilateral disease samples were also tested for differential expression against all samples from NC volunteers (using samples from subjects 1–4 as ‘true’ NC controls) which identified 24 significantly changed proteins (Appendix A). On comparison of these two discriminatory lists, three proteins (complement 8A, fibrinogen and plasminogen) were identified in both cases as significant to the BC samples (when compared with intra-individual controls and with ‘true’ NC controls). Nevertheless, differences were highlighted when using matched healthy samples from the same individuals and absolute NC samples as controls. However, for the most part, proteomes correlated highly between diseased and contralateral breast.

Paired samples were then categorized based on overall subject health status (NC or BC) for all further analyses rather than the contralateral ‘healthy’ control performing as an intra-individual baseline contrast for the ‘disease’ sample. Future models with larger sample sizing could further investigate these intraindividual variations, but this was beyond the scope of this study.

### 3.3. Quantification of Proteomic Pertubations in BC NAF Compared with NC

As the epithelial cells lining the breast ductal system undergo substantial changes in structure and function during breast cancer development, its analysis should form a basis for our general understanding of the effects of breast cancer on the NAF proteome profile. To determine whether significant proteins identified therein were representative of molecular changes in the surrounding tissues, proteomic comparison of NC and BC samples were explored for differential expression. This revealed a shift in the quantitative proteome composition. Of the 198 quantified protein groups, 40 had *p*-values ≤ 0.05, indicating a statistically significant difference in their abundance in cancer samples, compared to the NC NAF; 26 were classified as upregulated, and 14 classified as downregulated (Figure 2a) (Appendix A). Based on the relative expression profile of the significantly differential proteins, hierarchical clustering of samples grouped the majority of controls from disease (with the exception of one benign pair of samples), as many individual proteins showing opposite expression levels between NC and BC (Figure 2b).

Gene Ontology enrichment analysis of the altered BC proteome revealed the most significantly augmented biological process to be the immune response (*p* ≤ 0.01). Molecular function analysis revealed upregulated proteins to be associated with complement (component C9; component C8 alpha chain; complement factor B; C4b-binding protein alpha chain; complement component 3; complement decay-accelerating factor) (*p* ≤ 0.01) and protease inhibitor activity (kininogen-1; alpha-2-macroglobulin) (*p* ≤ 0.05) (Appendix A). The enriched molecular functions of downregulated proteins included cell adhesion molecule (desmocollin-2; lactadherin), receptor (polymeric immunoglobulin receptor; CD59 glycoprotein), complement (clusterin) and galactosyltransferase activity (beta-1,4-galactosyltransferase 1) (*p* ≤ 0.05) (Appendix A).

Enriched pathway analysis was performed by mapping the differentially abundant proteins to the Reactome pathway database, which identifies overrepresented pathways based on a Fisher’s exact test. The most significantly enriched pathways were those associated with the immune system, metabolism of proteins, extracellular matrix organization and hemostasis (Appendix A).

Immune response was consistently identified as a significant component of the NAF proteome by GO analysis, and its quantitative profile affected by BC, both by the upregulation and downregulation of protein groups when compared to NC data. As shown in Figure 3a, two major pathways of the innate immune system (blue bars) showed significant enrichment with 22 proteins involved (FDR = 1.06 × 10^−9^). Of those, the complement cascade was most significantly enriched involving 10 proteins. Seven components were upregulated, and particularly associated with the initial triggering (complement factor B; immunoglobulin heavy constant gamma 3; complement component 3) and regulation (complement decay-accelerating factor; component C9; complement factor B; immunoglobulin heavy constant gamma 3; complement component 3; C4b-binding protein alpha chain; component C8 alpha chain) of complement, by activation of complement component 3 and component 5 (complement component 3 upregulated). Downregulated proteins, CD59 glycoprotein, immunoglobulin heavy variable 3–33 and clusterin were associated with initial triggering and complement regulation. Many of these proteins (secretory leukocyte peptidase inhibitor, haptoglobin, alpha-1-B glycoprotein, CD55 molecule, S100 calcium binding protein A7, olfactomedin 4, CD59 molecule, polymeric immunoglobulin receptor and complement C3) are also associated with neutrophil degranulation (FDR = 2.49 × 10^−6^).

All differential proteins from the immune system were analyzed by STRING to assess confidence in their known protein–protein interactions (PPI), providing a PPI enrichment *p*-value of ≤ 1.0× 10^−16^ (Figure 3b).

Significantly abundant proteins were also associated with hemostasis (FDR = 6.10 × 10^−4^). For example, pathways of platelet activation, signaling and aggregation were enriched, in the upregulated BC dataset, involving proteins such as kininogen-1; plasminogen; alpha-1B-glycoprotein; fibrinogen gamma chain; serotransferrin; alpha-2-macroglobulin; three of which were also associated with the formation of fibrin clot (kininogen-1; fibrinogen gamma chain; mucin-1; alpha-2-macroglobulin). Proteins in the downregulated dataset were also enriched for these hemostatic pathways (clusterin; CD9 antigen; and mucin-1). As seen in Figure 3c, STRING analysis of the proteins showed a PPI enrichment *p*-value of ≤ 1.0 × 10^−16^.

Moreover, pathways of the extracellular matrix (ECM) organization were overrepresented in the significant dataset (FDR = 3.86 × 10^−2^), as upregulated proteins were associated with degradation of the ECM through activation of matrix metalloproteinase (plasminogen; alpha-2-macroglobulin) and integrin cell surface interactions (fibrinogen beta chain and fibrinogen gamma chain). As seen in Figure 3d, STRING analysis of these proteins showed a PPI enrichment *p*-value of 6.48 × 10^−8^.

Five upregulated (kininogen-1; plasminogen; fibrinogen gamma chain; serotransferrin and complement component 3) and one downregulated (lactadherin) protein were involved in metabolism of proteins by the regulation of insulin-like growth factor (IGF) transport and uptake by insulin-like growth factor binding proteins (IGFBPs) pathway (FDR = 1.23× 10^−4^). Plasminogen and fibrinogen are particularly important as precursors in the proteolysis of IGF, IGFBP and acid-labile subunit ternary complex, thus releasing the IGF. As seen in Figure 3e, STRING analysis of these six proteins showed a PPI enrichment *p*-value of ≤ 1.0× 10^−16^.

## 4. Discussion

There is an unmet clinical need for increased monitoring of breast health, particularly in high-risk women, to detect breast cancer earlier. An alternative approach to the established imaging modalities is required. NAF is a unique liquid biopsy that can be used to assess molecular changes associated with breast cancer progression [22]. The original proteomics approach was time-consuming and in order to validate biomarkers linked to breast cancer, a more rapid method is required for clinical trials recruiting larger cohorts. The aim of this study was to demonstrate the proof of principle that a rapid LC MS method could be used to study the most abundant proteins in NAF and that changes in profile correlated with disease.

All NAF samples collected in this study were expressed using breast massage techniques only, with approximately 50% success rate. Various methods have been developed to increase yield, using chemical (90% success rate with administered oxytocin) [15,28] or mechanical (70% using a breast pump/syringe) intervention [29,30]. However, by only using massage techniques, we can assess the practicalities of implementing NAF collection as a surveillance strategy starting in a domestic environment.

Samples were characterized by volume acquired, color and visible viscosity as a significant association between the color of NAF and breast cancer development has previously been observed. Red/brown NAF was reported to be more common in breasts with ductal carcinoma in situ than atypical hyperplasia (*p* = 0.008). The predictive model, that included NAF color, cytology, and age, was 92% sensitive and 61% specific in predicting breast cancer status [31]. During our analysis, we identified one sample from the diseased breast of a patient with invasive carcinoma to be of a red color (BC7), whilst the NAF of healthy breast origin was described as yellow; all other IC samples were labelled as brown, yellow or white with one olive-green sample being an exception. The color of NAF has since been strongly linked with dietary intake, and a red/brown discoloration being attributed to lipids levels (mainly as oxidized cholesterol end-products) and green discoloration linked with an absence of lactose intake [32].

While the NAF proteome has been reported to share considerable commonalities with the plasma proteome as defined by highly abundant proteins, and by dominant bands in the SDS-PAGE profiles, NAF differed from plasma in that the specific proteins that dominated, varied between individuals and within matched samples of unilateral disease. Immunodepletion, which is often used in serum and plasma proteomics to remove most abundant proteins, was avoided with NAF to minimize experimental variation in sample preparation. Although there is concern of more abundant and dynamic proteins masking the detection of less abundant candidates from the surrounding disease tissue, our 1D LC-MS/MS analysis demonstrated sufficient resolution for detecting relative changes in protein abundance in response to BC. A shift in the quantitative proteome composition was identified in BC NAF, as the relative abundance of 40 proteins were significantly different. The variation in most abundant protein levels prevents any one component being used as an internal standard to improve quantitative accuracy. Future development of NAF 1D LC-MS/MS proteomics profiling will incorporate a spiked-in quantitative standard protein such as yeast alcohol dehydrogenase. Further refinements of scale and preparation steps can be made by reducing the amount of initial protein used from 200 μg, as only 2 μg is required for each LC-MS/MS analysis. Hence there is scope for significant reduction in the scale and number of steps of sample preparation.

The objective of this study was to demonstrate the proof of principle that a rapid LC-MS/MS method could be used for the study of the most abundant proteins in NAF and that changes in profile correlated with disease. Due to the small sample numbers available for this pilot study, we were unable to identify specific biomarkers or to assess potential con-founding factors such as age or reproductive history. Nevertheless, changes in protein clusters provided interesting insights in disease progression and differentiation of NC from BC. A prominent imbalance in immune response representatives were identified between NC and BC groups. The innate immune system has significant importance in the initiation and progression of cancer [33,34]. Complement cascade activation is involved in both the development of and defense against cancer and carries exceptionally harmful pro-inflammatory potential. Upon activation, complement proteins mediate several effector functions such as inflammatory cell and fibroblast recruitment to the tumor microenvironment, which stimulates extracellular matrix remodeling and consequently supports cancer progression [35]. Of those differentially expressed in BC, the complement cascade was most significantly enriched for 10 factors. Seven of these components, significantly more abundant in cancer NAF, were associated with the initial triggering (complement factor B; immunoglobulin heavy constant gamma 3; complement component 3) and regulation (complement decay-accelerating factor; component C9; complement factor B; immunoglobulin heavy constant gamma 3; complement component 3; C4b-binding protein alpha chain; component C8 alpha chain) of complement, by activation of complement component C3 and component C5 (complement component 3 upregulated).

It is believed that during the first stages of breast cancer development, the innate immune system acts as a key player in tumor immunoediting and is the host’s defense against cancer. Panels of soluble molecules can rapidly detect abnormal cells, in which the innate immune system response targets and orchestrates an attack against disease progression through proteins of the complement cascade and cytokine signaling. However, there is an increase in emerging literature reporting involvement of the complement cascade in several hallmarks of cancer, as it can also be a part of the long-lasting inflammatory status that can lead to malignant transformation and tumor development [36]. It has been hypothesized that the presence of both complement activation products such as those notably detected in NAF samples (complement factors) is a host defense mechanism against cancer, and cancer cells resist complement attack by overexpressing complement regulatory proteins [37]. The regulation of complement pathway was also enriched and the complement regulatory protein that protects epithelial cancer cells from complement attack, CD59 glycoprotein (protectin) was detected in cancer NAF samples [38].

In addition to the immune/inflammatory response and elements of the breast microenvironment, the close relationship between breast tumors and the hemostatic system is recognized as an important regulator of breast cancer progression. Elements of the hemostatic system, such as platelets, coagulation, and fibrinolysis, can have a direct impact on many of the reported hallmarks of breast cancer and related closely to the immune response system. These include mechanisms of cellular transformation, sustained proliferation and cell survival, prevention of apoptosis, angiogenesis and tissue invasion and metastasis [39]. Of the differentially expressed proteins characteristic to BC samples, nine had associated roles in the platelet activation, signaling and aggregation pathway. Platelets function as exocytotic cells, secreting a multitude of effector molecules upon activation, usually at sites of vascular injury. Proteins associated with degranulation were upregulated in BC. For instance, kininogen-1; plasminogen; alpha-1B-glycoprotein; fibrinogen gamma chain; serotransferrin; alpha-2-macroglobulin involved in exocytosis of platelet alpha granule contents [40]. Platelet alpha granules release many factors which can further enhance breast cancer cell adhesion, proliferation, chemotaxis, and chemo invasion via several mechanisms [41]. For example, plasminogen which is converted to plasmin by serine proteases (such as tissue-type plasminogen activator) degrades the extracellular matrix to assist in breast carcinoma invasion and progression and was upregulated [42]. Components of this pathway, such as tissue plasminogen activator, are actually used as breast cancer biomarkers for monitoring therapy and have been reported in other NAF studies [43]. Many of the proteins identified as significantly changed in NAF, connected to immune and inflammation responses, are not uniquely associated with BC, or cancer in general *per se*. The striking value of our study is that if the changes were observed in serum/plasma they would be rendered non-specific, unassociated with a specific tissue or organ or disease. However, detection of variations in NAF are breast-enriched due to the proximity of the liquid produced, to the cells which are the origin of the majority of breast cancers.

## 5. Conclusions

Developments in proteomics have empowered the analysis of hundreds of proteins, permitting the identification of new biological markers. By profiling patient samples and healthy controls, we can begin to understand the differential expression of proteins at the earliest stages of breast cancer development. The enormous potential NAF carries as a liquid-biopsy for this purpose was highlighted in this preliminary study, as samples were obtained readily by manual massage techniques. BC was reflected in both intra-individual (healthy/disease bilateral) and inter-individual (NC/BC) NAF proteomes as candidate proteins, representative of their changing microenvironment, were identified with varying abundance. These proteins encompassed a variety of altered molecular functions that have previously been reported as associated with development of the disease, as well as those involved in specific biological pathways. Proteome differences were measured without extensive sample preparation, thereby improving throughput and making the prospect of using NAF in a clinical setting much more feasible.

With this modified workflow for characterizing NAF, we now have the potential to study a larger cohort of volunteers and patients. This will enable (a) further investigation of biomarkers indicative of earlier stages of breast cancer, (b) further investigation of using the contralateral non-diseased breast from the same individual as a control for individual heterogeneity and (c) assessment of confounding factors that contribute to NAF proteome variation such as menopausal status and diet.

## Figures and Tables

**Figure 1 jcm-10-02243-f001:**
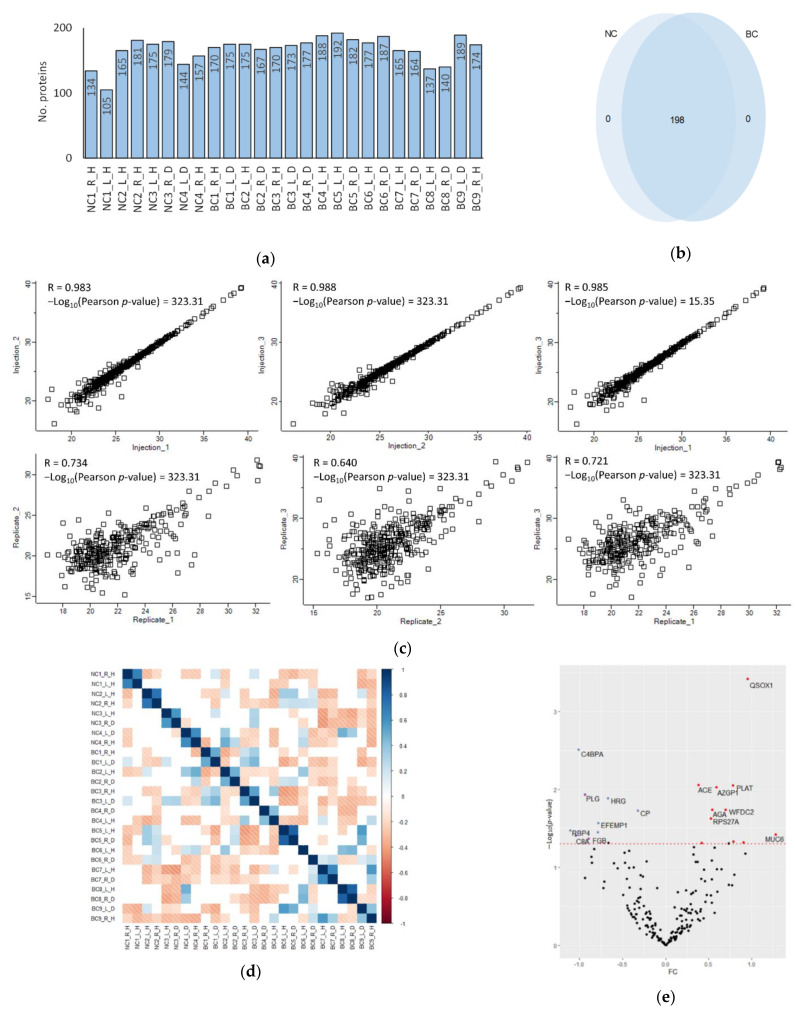
Assessment of NAF sample proteomes. (**a**) Number of proteins identified in each sample and (**b**) comparison of BC proteomes with NC; (**c**) Scatter plot comparison represents Pearson correlation coefficient (R) of Log_2_ annotated protein groups from biological (Replicate_1,2,3) and technical replicates (Injection_1,2,3) of subject 9 right breast sample (**d**) Pearson correlation matrix between all samples (positive correlations = blue, negative correlations = red, non-significant correlations = white); (**e**) Volcano plot of protein fold-change between bilateral samples (disease breast versus the contralateral non-diseased breast sample) from BC patients plotted against paired student *t*-test *p*-value. Red dashed line shows a *p*-value cut off = 0.05; (red spots = upregulated proteins, blue spots = downregulated proteins); (right/left = R/L, healthy/disease = H/D).

**Figure 2 jcm-10-02243-f002:**
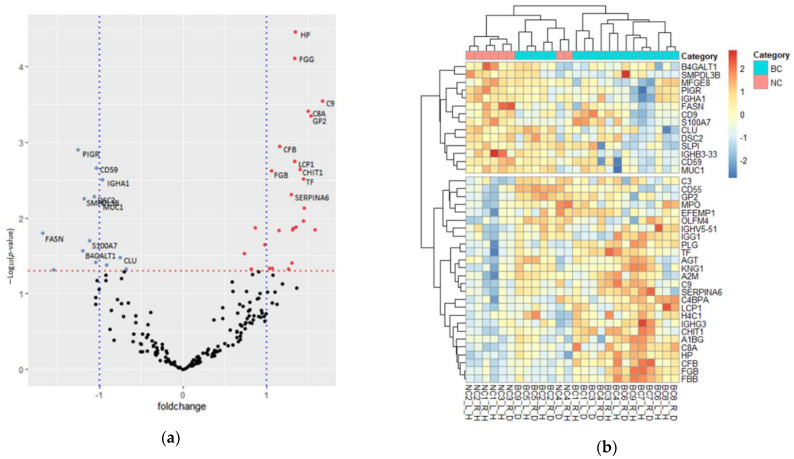
Differentiated protein expression between NC and BC NAF (**a**) Volcano plot of protein fold-change across the NC and BC NAF samples plotted against the unpaired student *t*-test *p*-value. Red dashed line shows a *p*-value cut off = 0.05; (red spots = upregulated proteins, blue spots = downregulated proteins). (**b**) Hierarchical clustering of proteins differentially expressed between NC (pink) and BC (blue), plotted as a z-score for each sample (red = up, blue = down) (right/left breast = R/L, healthy/disease breast = H/D between bilateral samples).

**Figure 3 jcm-10-02243-f003:**
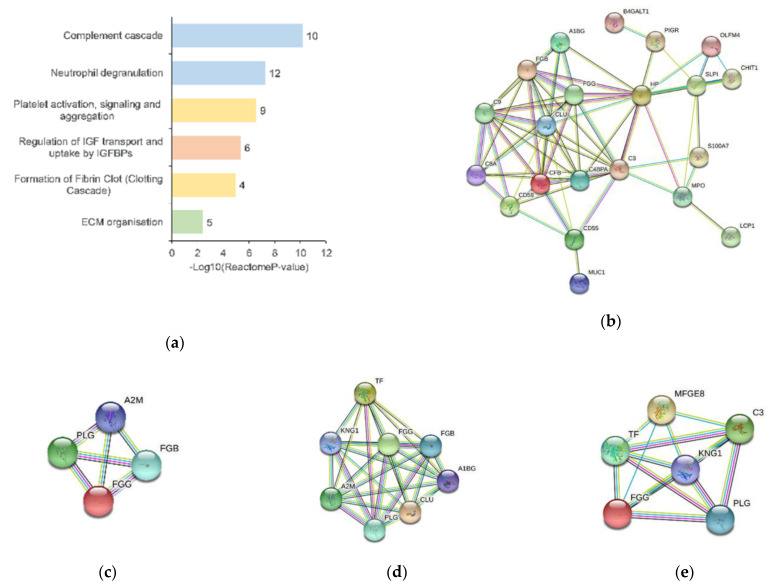
Pathway analysis of differential proteins in BC NAF. (**a**) Reactome pathway analysis of differential expressed proteins of BC NAF. Significantly enriched pathways included the innate immune system (blue), metabolism of proteins (orange), hemostasis (yellow) and ECM organization (green) (*p* ≤ 0.05). Each pathway is labelled with the number of significantly differentiated proteins involved in the listed pathway; the STRING protein database was used to analyze the protein–protein interactions of significantly differentiated protein associated with (**b**) the immune system pathways; (**c**) ECM-associated proteins; (**d**) hemostatic proteins; (**e**) proteins associated with the IGF pathway; colored lines between the nodes (proteins) indicate the various types of interaction evidence in the STRING protein database.

**Table 1 jcm-10-02243-t001:** Clinical parameters of NC (*n* = 4) and BC (*n* = 9) subjects selected for NAF LC-MS/MS analysis.

Subject	Pathology	Disease Breast	Grade	TNM Staging	Phenotype	Age	Pre/Post-Menopausal	Breast	NAF—Appearance	Conc. mg/mL	Total Protein (μg)
NC1	Normal	N/A	N/A	N/A	N/A	48	Pre	Left	White, cloudy	15.86	793.00
								Right	White, cloudy	18.20	910.00
NC2	Normal	N/A	N/A	N/A	N/A	31	Pre	Left	Light brown-waxy	34.72	4999.49
								Right	Light brown-waxy	33.30	4961.17
NC3	Benign (phyllodes)	Right	N/A	N/A	N/A	43	Pre	Left	White, cloudy	16.53	793.00
								Right	Light brown, cloudy	21.50	1892.00
NC4	Benign (duct ectasia)	Left	N/A	N/A	N/A	59	Post	Left	Terracotta, thick	65.81	7831.75
								Right	Light brown, thick	81.20	8039.20
BC1	DCIS	Left	N/A	N/A	N/A	62	Post	Left	Yellow, cloudy	26.80	750.00
								Right	Yellow, cloudy	27.52	770.00
BC2	DCIS	Right	N/A	N/A	N/A	57	Post	Left	Light brown, cloudy	63.25	1138.46
								Right	Light brown, cloudy	28.04	504.79
BC3	IC (ductal)	Left	1	G1 T3 N?	ER+ve/HER2−ve	59	Post	Left	White, cloudy	3.28	522.00
								Right	Olive green, thick	15.82	1725.00
BC4	IC (ductal)	Right	3	G3 pT2 pN0	ER+ve/HER2−ve	50	Post	Left	Light brown, cloudy	35.14	1687.00
								Right	Light brown, cloudy	22.74	637.00
BC5	IC (ductal)	Right	3	G3 T1c or T2 N0?	ER+ve/HER2−ve	86	Post	Left	Light brown, cloudy	58.81	11,643.43
								Right	Light brown, cloudy	69.48	8893.00
BC6	IC (ductal)	Right	1	pT2	ND	48	Pre	Left	White, cloudy	14.65	1743.25
								Right	Light brown, cloudy	42.30	3342.02
BC7	IC (ductal)	Right	ND	pT1c N1	ND	47	Pre	Left	Yellow, translucent	35.46	921.96
								Right	Red, opaque	53.80	699.40
BC8	IC (ductal)	Right	1	G1 ypT1c ypN2	Tubular	44	Pre	Left	White, opaque	39.72	993.00
								Right	White, opaque	22.13	774.55
BC9	IC (ductal)	Left	2	G2 pT1c	NST	47	ND	Left	Yellow-brown, thick	46.91	3471.64
								Right	Yellow-brown, thick	62.97	7493.91

NC—non-cancer, BC—breast cancer, DCIS—ductal carcinoma in situ, IC—invasive carcinoma, N/A—Not applicable, G—Grade (1–3), T—Tumor status, N—Lymph node status, *p*—pathological stage, ?—Undefined, ER—Estrogen receptor status, HER2—Human epidermal growth factor receptor 2 status, +ve—Positive, −ve—Negative, ND—Not disclosed.

## Data Availability

The data supporting the findings of this study are available within the Appendix A.

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
