# Peer review of "High-Throughput Proteomic Profiling of Nipple Aspirate Fluid from Breast Cancer Patients Compared with Non-Cancer Controls: A Step Closer to Clinical Feasibility"

_jcm, 2021, doi:10.3390/jcm10112243_

Round 1

Reviewer 1 Report

After revision, the manuscript has been improved. I have no further comments.

Author Response

The authors would like to thank the reviewer for their feedback on the completion of the manuscript.

Reviewer 2 Report

High-throughput proteomic profiling of nipple aspirate fluid from breast cancer patients compared with non-cancerous controls: a step closer to clinical feasibility

This is a nicely described pilot study that aims to assess the utility of a rapid LC-MS approach for screening nipple aspirate fluid for potential biomarkers of breast cancer. This would reduce the need for invasive screening techniques for women with inconclusive mammograms. The authors have avoided extensive sample preparation in order to make the assay more feasible in the clinical setting. Whilst in depth proteome coverage has been sacrificed, the authors aimed to determine whether variations in the higher abundance proteins could still provide meaningful patient stratification. The number of patient samples included in the pilot study were understandably small. This together with the very variable nature of nipple aspirate fluid makes confident interpretation of the results rather challenging. Nonetheless, the authors have shown that a rapid screening approach may have some potential.

A few comments. The authors showed that there was little difference between the proteomes of the nipple aspirate fluid from healthy and diseased breast from the same donor, and the contralateral ‘normal’ fluid grouped with BC samples in their analyses. However, their hypothesis is that the nipple aspirate contains useful diagnostic proteins because they are derived from the local microenvironment of the diseased tissue. Can they therefore postulate why the proteomes are the same between the bilateral samples? Did the authors analyse any of the other factors that might feasibly result in changes to the nipple aspirate, such as menopausal status?

Not all of the terms and/or abbreviations are sufficiently clear. The authors have used NC and BC for non-cancer and breast cancer, respectively. However, in figures 1 and 2, samples are labelled HV and BN, which I have assumed means healthy volunteer and benign – the labels need to be clarified and consistent. Also, the only place I can see where NC is defined as both normal (HV?) and benign (BN), and BC as both DCIS and IC, is in the heading for Table 1. I think this should be included very clearly in the text.

Some of the fold changes listed in Supplementary tables 1 and 2 are very small, indeed in one case the fold change is 1 but it has a P value of 0.023 (Supplementary table 1b, protein P62805). This makes me slightly concerned about the statistical approach that was used. Could the authors comment? Would a scatterplot of normalised intensities be more revealing? The heatmap in figure 2b shows the samples ‘in order of their respective disease groups.’ I would prefer to see them subjected to hierarchical clustering as a more objective way of visualising differences between the clinical groups.

Stylistically, there are many abbreviations used in this paper which occasionally makes sections difficult to read. It would be more digestible if the authors could limit these somewhat. In addition, not all of the abbreviations have been defined – TNM, PSMs and quite a few of the protein names (KNG1, DSC2, MFGE8 etc), plus the TNM staging in table 1 has also not been defined. I personally would prefer the protein names as well as the accession numbers to be included in the supplementary tables as again this is more instantly digestible for the reader. The title of the paper could be a little punchier.

Some minor typos:

Page 1, line 38 – of not to national screening programs.

Page 6, line 2 – an unnecessary or before 1 in 20.

Page 6, line 18 – ‘A second step digest was applied…’ I’m not sure what this means – the protein was diluted before further incubation?

Page 6, line 24 – Tribrid not Tribid.

Page 6, line 43 – there are in the region of 20,000 human proteins, I believe the number 562,118 refers to the total number of entries in SwissProt.

Page 6, line 47 – do you mean deamidation rather than deamination?

Page 7, line 95 – right breast not beast.

Page 9, line 123 – ‘Consequently, for this study…’ This sentence isn’t clear and is quite important, so please could you rephrase it.

Page 10, line 171 – ‘These proteins are also implicated…’ This sentence is also unclear.

Figure 3 – there is no key for b)-e) so I’m not sure how to interpret these images.

Author Response

Reviewers' Comments to the Authors:

Reviewer #2: “This is a nicely described pilot study that aims to assess the utility of a rapid LC-MS approach for screening nipple aspirate fluid for potential biomarkers of breast cancer. This would reduce the need for invasive screening techniques for women with inconclusive mammograms. The authors have avoided extensive sample preparation in order to make the assay more feasible in the clinical setting. Whilst in depth proteome coverage has been sacrificed, the authors aimed to determine whether variations in the higher abundance proteins could still provide meaningful patient stratification. The number of patient samples included in the pilot study were understandably small. This together with the very variable nature of nipple aspirate fluid makes confident interpretation of the results rather challenging. Nonetheless, the authors have shown that a rapid screening approach may have some potential.”

Major concerns

  • The authors showed that there was little difference between the proteomes of the nipple aspirate fluid from healthy and diseased breast from the same donor, and the contralateral ‘normal’ fluid grouped with BC samples in their analyses. However, their hypothesis is that the nipple aspirate contains useful diagnostic proteins because they are derived from the local microenvironment of the diseased tissue. Can they therefore postulate why the proteomes are the same between the bilateral samples? Did the authors analyse any of the other factors that might feasibly result in changes to the nipple aspirate, such as menopausal status?

Response: The authors would like to thank the reviewer for their comment regarding similarity of proteome in the same individual versus the micro-environmental protein changes. We have previously postulated that cross-lymphatic drainage between the breast may contribute an equilibrium in the proteomes (Shaheed 2017). There is a possibility that there are differences in the levels of individual proteins that differentiate the disease from the contralateral healthy breast, and the investigation of this would be one of the objectives of a larger study.

Equally, it would have been interesting to explore confounding factors on the NAF proteome, such as menopausal status. However, in our study, this would not be possible because of limited sample sizing and would be answered in a larger study once we have an established method with sufficient throughput. We have modified the conclusion to reflect the additional understanding of these parameters that will be gained from a larger study. 

  • Not all of the terms and/or abbreviations are sufficiently clear. The authors have used NC and BC for non-cancer and breast cancer, respectively. However, in figures 1 and 2, samples are labelled HV and BN, which I have assumed means healthy volunteer and benign – the labels need to be clarified and consistent. Also, the only place I can see where NC is defined as both normal (HV?) and benign (BN), and BC as both DCIS and IC, is in the heading for Table 1. I think this should be included very clearly in the text.

Response: We agree with the reviewer’s assessment. As we do not make any conclusions from these differentiated sub-groups (HV/benign) in our figures, we have re-labelled the images to clearly show non-cancer subjects 1-4, and breast cancer subjects 1-9. We have now also clearly defined additional abbreviations of right or left, healthy or disease breast in the figure legends.

  • We have used consistent labels (NC 1-4 or BC 1-9) within Table 1 when describing the clinical parameters of each subject
  • Relabelled figures 1a, 1d and 2b

  • Some of the fold changes listed in Supplementary tables 1 and 2 are very small, indeed in one case the fold change is 1 but it has a P value of 0.023 (Supplementary table 1b, protein P62805). This makes me slightly concerned about the statistical approach that was used. Could the authors comment? Would a scatterplot of normalised intensities be more revealing?

Response: We would like to thank the reviewer for pointing this out. The values reported were in fact Log2(fold change) values. The supplementary table headings have now been corrected.

  • The heatmap in figure 2b shows the samples ‘in order of their respective disease groups.’ I would prefer to see them subjected to hierarchical clustering as a more objective way of visualising differences between the clinical groups.

Response: As suggested by the reviewer, we have now ordered the samples in Figure 2b heat map as subjected to hierarchical clustering.

  • Stylistically, there are many abbreviations used in this paper which occasionally makes sections difficult to read. It would be more digestible if the authors could limit these somewhat. In addition, not all of the abbreviations have been defined – TNM, PSMs and quite a few of the protein names (KNG1, DSC2, MFGE8 etc), plus the TNM staging in table 1 has also not been defined. I personally would prefer the protein names as well as the accession numbers to be included in the supplementary tables as again this is more instantly digestible for the reader.

Response: We have limited the use of abbreviations where possible to make the paper more digestible for the reader. All abbreviations have now been clearly defined, and where we have used gene names in the main text, we have replaced with full protein names. Within the supplementary tables, all lists have been updated to include accession numbers, gene name and full protein names.

Minor Concerns:

Response: Thank you for highlighting these errors and areas requiring further clarification. We have amended the manuscript and all changes have been highlighted.

Page 1, line 38 – of not to national screening programs.

Page 6, line 2 – an unnecessary or before 1 in 20.

Page 6, line 18 – ‘A second step digest was applied…’ I’m not sure what this means – the protein was diluted before further incubation?

Page 6, line 24 – Tribrid not Tribid.

Page 6, line 43 – there are in the region of 20,000 human proteins, I believe the number 562,118 refers to the total number of entries in SwissProt.

Page 6, line 47 – do you mean deamidation rather than deamination?

Page 7, line 95 – right breast not beast.

Page 9, line 123 – ‘Consequently, for this study…’ This sentence isn’t clear and is quite important, so please could you rephrase it.

Page 10, line 171 – ‘These proteins are also implicated…’ This sentence is also unclear.

Figure 3 – there is no key for b)-e) so I’m not sure how to interpret these images.

Response: We have corrected all minor concerns highlighted by the reviewer.

This manuscript is a resubmission of an earlier submission. The following is a list of the peer review reports and author responses from that submission.

Round 1

Reviewer 1 Report

The aim of this study was to evaluate the utility of a 1D proteomics method in the identification of differential protein abundances in nipple aspirate fluid between breast cancer and control subjects. This is exciting as it would facilitate larger studies on this topic and support the development of a noninvasive breast cancer screening tool. However, I have some comments that should be addressed in order to improve the quality of the study.

Major: The statistical methods are not described with sufficient detail for the differential protein analyses, both for the within-subject analysis and the BC vs NC analysis

  • Was the t-test used to identify differences between the diseased and normal breast a paired t-test? (Line 111)
  • What was the statistical model used for the identification of differential protein abundance between NC and BC subjects?
  • Were potential confounding factors such as age considered, and was the model adjusted for these?
  • How was the correlation between a subject's left and right sample accounted for in the model? 
  • The usage of a unadjusted p-values should be justified somewhere
  • Supplemental tables should show at least 2 significant figures for effect sizes and p-values. P-values should not be "0.0" but scientific notation should be used to show precisely the p-value

Minor: Figure 4e is incorrectly referred to as Figure 3e (Line 187)

Reviewer 2 Report

In this paper, the authors optimize a new methodology to characterize the proteomic profiles of nipple aspirate fluid (NAF) in early breast cancer patients. The applied approaches represent a very good example of the needed new approaches and methodologies in the liquid biopsy field, specifically in breast cancer. Applying the poorly studied proteomic analyses in liquid biopsy together with NAF samples that are potentially better reflecting the “omic” profiles of these tumours, paves the way to new concepts and knowledge in this fascinating field.

Despite the study is presented as a proof-of-concept, I found several caveats that need to be solved before publication.

Major comments

  1. The data set is composed by 9 samples from breast cancer patients and 4 healthy controls. Considering the breast cancer prevalence in occident, more cases are needed in order to present consistent and significative results as the authors specified in lines 333-334: “With this higher-throughput workflow, sample sizes can be increased in future NAF proteomic analysis to statistically validate these differences as reliable candidate biomarkers”. This workflow needs to be validated in a higher cohort of patients.
  2. Figure 1. This figure represents just a control staining for protein presence without interest to be main figure. It should go into supplementary materials.
  3. Figure 2b. The number of proteins belonging exclusively to each condition has to be indicated in the diagram.
  4. Figure 2c. The same analysis has to be performed for all samples from all subjects and at least showed in the supplementary material. P-values should be indicated.
  5. Figure 2d. It is not clear what is being measured in the matrix and what the heatmap means.
  6. Figure 2e. The authors should apply this analysis with each disease sample versus their bilateral healthy counterpart as well as each disease sample versus all healthy samples.
  7. Figure 3b. This part of figure 3 is missing in the main text and not properly explained.
  8. The discussion should be summarized. Most of the information belongs to introduction.

Minor comments

  1. Figure 3 description is misspelled. “Figure 3. Differential protein analysis between NC and BC NAF (a) Volcano plot of protein Figure 14. (blue dots) classified as downregulated”

Reviewer 3 Report

Currently, multiple studies are ongoing with the aim to better define the diagnostic options, mostly in relation to the prognostic factors for breast cancer. 

It could be of interest to add a brief description of the different breast molecular subtypes in terms of impact on prognosis and how radiomics is trying to predict on the base of imaging tumor characteristics.   It could be of interest to cite 
  • PMID: 2913256 - DOI: 10.1016/j.soc.2017.08.005
  • PMID: 33299040 - DOI: 10.1038/s41598-020-78681-9

Moreover, it could be of interest to discuss the possible impact on worldwide screening system or how this technique could be integrated with standard imaging.

Round 2

Reviewer 1 Report

After revision, the manuscript has been improved and I have no further comments. 

Author Response

The authors thank the reviewer for accepting the changes to the manuscript.

Reviewer 2 Report

The approach is highly novel and interesting to the field. The paper has been improved; however, the number of samples are not sufficient to support the results described and lack real interest. The limitations in the current pandemic situation are understandable although sample obtention doesn´t feel as problematic as the authors stated. 

Some figures are still confusing and not very well explained.

Author Response

The authors would like to thank the reviewer for their helpful suggestions; -

  • The number of samples are not sufficient to support the results described and lack real interest. The limitations in the current pandemic situation are understandable although sample obtention doesn´t feel as problematic as the authors stated. 

Response. Due to the pandemic the breast oncoplastic surgeons at Bradford Teaching Hospitals NHS Foundation Trust with who we collaborate have had to prioritise COVID-19 patient care, dealing with a 20% morality rate due to the high deprivation level in the region. The clinicians have significantly reduced clinics and theatre sessions to treat breast cancer, much of which has been moved to a different hospital. It has become a major national concern that the full range of breast screening methods and treatments have had to be reduced and this could lead to increased mortality from breast cancer. Furthermore, the recruitment and ethical consenting of volunteers has been restricted due to lockdown and social distancing preventing our team from engaging with patients or healthy individuals. This inevitably has led to the inability to gain further samples for the study.

We have endeavoured to make it clear that this is a pilot study throughout the manuscript. We have carefully avoided the claim to identify specific biomarkers because of the size of the study. By demonstrating the proof of principle of the technique, we hope to gain further investment into the project so that we can recruit dedicated clinicians and research staff for the recruitment of volunteers and gather more samples for a more extensive study, once the pandemic is over.

  • Some figures are still confusing and not very well explained.

Response. We have added additional details to the legends of the figures to improve clarity of the data contained therein.